# ICFI: A Feature Importance Measure for Multi-Class Classification

## Abstract

Feature importance is one of the most prominent methods in eXplainable Artificial Intelligence (XAI). It aims to assess the extent to which a machine learning model relies on different features. However, in multi-class classification, current methods fail to explain inter-class relationships, either because they provide explanations for binary classification only, or because they suffer from aggregation bias. To address these shortcomings, we propose Inter-Class Feature Importance (ICFI), which provides feature importance scores for discriminating between an arbitrary pair of classes. ICFI is a post-hoc, model-agnostic method, which provides bounded scores for interpretability. We empirically demonstrate through extensive experiments on real-world datasets that ICFI effectively captures the discriminating features between class pairs, outperforming existing methods.

## 1 Introduction

EXplainable Artificial Intelligence (XAI) focuses on making Machine Learning (ML) models understandable to stakeholders (König et al., 2021). XAI aims to address issues of ML model complexity in complying with legal requirements (Tritscher et al., 2023; König et al., 2021; European Parliament & Council of the European Union), validating architectures in high-stake scenarios (Dinu et al., 2020), and treating protected groups fairly (Caton & Haas, 2024). Feature Importance (FI) is one of the most popular XAI methods (Saarela & Jauhiainen, 2021), quantifying relevance of input features for model prediction (Muschalik et al., 2023), allowing analysis of alignment with background knowledge (Alfeo et al., 2023).

Current FI methods typically provide explanations for single instances (locally), or explain the entire data and model (globally), but for binary classification only, and cannot capture inter-class relationships. In particular, while an overall importance scoring stands valid for a certain percentage of the instances, it may not be accurate for all; an effect known as aggregation bias (Mehrabi et al., 2022).

For example, in kidney cancer detection (Muhamed Ali et al., 2018; Cancer Genome Atlas Research Network et al., 2013), clinical data and RNA sequencing information are used to detect cancer sub-types, constituting a multi-class classification task. Global FI provides a single ranking, quantifying feature contributions, but does not consider class relationships. Certain features might be important to separate two specific cancer sub-types while being otherwise less relevant for other classes. Existing FI methods fail to capture this. Moreover, a feature crucial for differentiating between two cancer types may not be found globally important, which risks critical oversights. Another case is a black-box model supporting diagnosis of patients with pneumonia, the flu, or a cold based on observed symptoms. Existing XAI methods explain why a patient was diagnosed with e.g. pneumonia, but do not offer information about the difference to other possible diseases with overlapping symptoms. Here, ICFI offers vital insights on the differences to other diagnoses, such as which of the observed symptoms distinguish pneumonia from the flu (Bowen & Ungar, 2020).

Consider e.g. Confusion Matrix (CM) pairwise class false positives and negatives (Beauxis-Aussalet & Hardman, 2014). From its entries, we might notice that a model does not distinguish between classes equally well. How do we understand why this happens? Existing FI methods only provide feature importance values globally across all classes, and cannot answer this question. Even worse, their aggregation of sample importance regardless of their class may be misleading in describing what features are of importance for a particular pair of classes that where confusion entries are high.

To address this gap, we propose a new feature importance measure to explain an ML model in multi-classification: Inter-Class Feature Importance (ICFI). ICFI generalizes FI approaches for binary classification and scores features for any pair of classes the user is interested into, quantifying feature importance in separating specifically those two classes. ICFI thus allows the user to inspect precisely those pairs of classes that they are interested in, e.g. to understand their differences generally speaking, to scrutinize how or why the model can or cannot differentiate them, or to identify descriptive features between them. Please note that existing FI measures applied to the classes of interest does not achieve this goal. They assess the importance of features for the overall classification task, not capturing relationships between particular classes. Moreover, even if one were to train separate binary classifiers specifically for the two classes of interest, FI values obtained for this separate model do not necessarily reflect the FI values of the original multi-class model (cf. also our empirical study, Sect.4). ICFI supports the analysis of any existing ML model in a post-hoc, model-agnostic manner, without having to (re-)train the model. In order to quantify feature importance, we adopt a common perturbation based strategy to mimic absence of features from the model (Covert et al., 2021), and demonstrate how it can be leveraged to provide pairwise class information.

Our contribution includes the introduction of the problem of pairwise feature importance in multi-class classification, the presentation of ICFI, a model-agnostic post-hoc feature importance method that offers an attractive model-agnostic post-hoc solution to this problem, discussion of ICFI's properties and use cases. Through empirical evaluation of real-world datasets, we demonstrate the reliability and effectiveness of ICFI in addressing the pairwise feature importance problem.

## 2 RELATED WORK

XAI is a dynamic research field; cf. recent surveys (Ali et al., 2023; Theissler et al., 2022; Das & Rad, 2020). *Model-specific* approaches are tailored to a specific model type only (Carletti et al., 2023; de Sá, 2019; Sundararajan et al., 2017; Bach et al., 2015), whereas *model-agnostic* ones are not. *Post-hoc* methods target fully trained models (Ali et al., 2023).

In our work, we focus on feature importance (FI) methods which score importance of model features (Das & Rad, 2020). SHAP (Lundberg & Lee, 2017) provides local FI only for binary classification via probabilities on one of the classes, adopting Shapley Values from game theory, and suffers from high computational complexity (Muschalik et al., 2023). Global SHAP explanations average local ones. SHAP does not take into account inter-class relationships and suffers from aggregation bias, i.e., misrepresents samples whose importance does not align with the average (Mehrabi et al., 2022). GSHAP generalizes traditional SHAP explanations by computing Shapley Values on any function of the model output instead of the model output itself Bowen & Ungar (2020).One of the GSHAP methods studies which features distinguish a pair of classes, thus studying inter-class feature importance. However, this requires defining a positive and a negative class. The resulting formulation suffers from the difficulty that it is not symmetric and the FI depends on which class has been chosen as the positive class, potentially providing contrasting explanations. Moreover, relying on SHAP, GSHAP suffers from the same high computational complexity that characterises SHAP values. LIME (Ribeiro et al., 2016) provides local FI by approximating the model locally with an interpretable surrogate (Adamczewski et al., 2020; Ribeiro et al., 2016). Global FI is obtained by aggregating local ones (Ribeiro et al., 2016). LIME suffers from instability to changes in the surrogate's input (Zhou et al., 2021).

Global XAI approaches include Partial Dependence Plots (PDP) (Friedman, 1991) and Permutation Feature Importance (Breiman, 2001). PDPs are a low-dimensional visualization of the dependence between a target variable and a set of features of interest (Greenwell et al., 2017). PDPs do not target FI but rather the interaction between the target and a set of input features. PFI, introduced in Breiman (2001) for random forests, assesses change in the model's performance when permuting the feature of interest to mimic its absence, effectively marginalizing the other features (Fumagalli et al., 2023; Strobl et al., 2008). While PFI has been originally introduced for tree models only, it has recently been expanded to be model-agnostic Fisher et al. (2019); Fumagalli et al. (2023), but is limited to purely global settings, potentially suffering from aggregation bias.

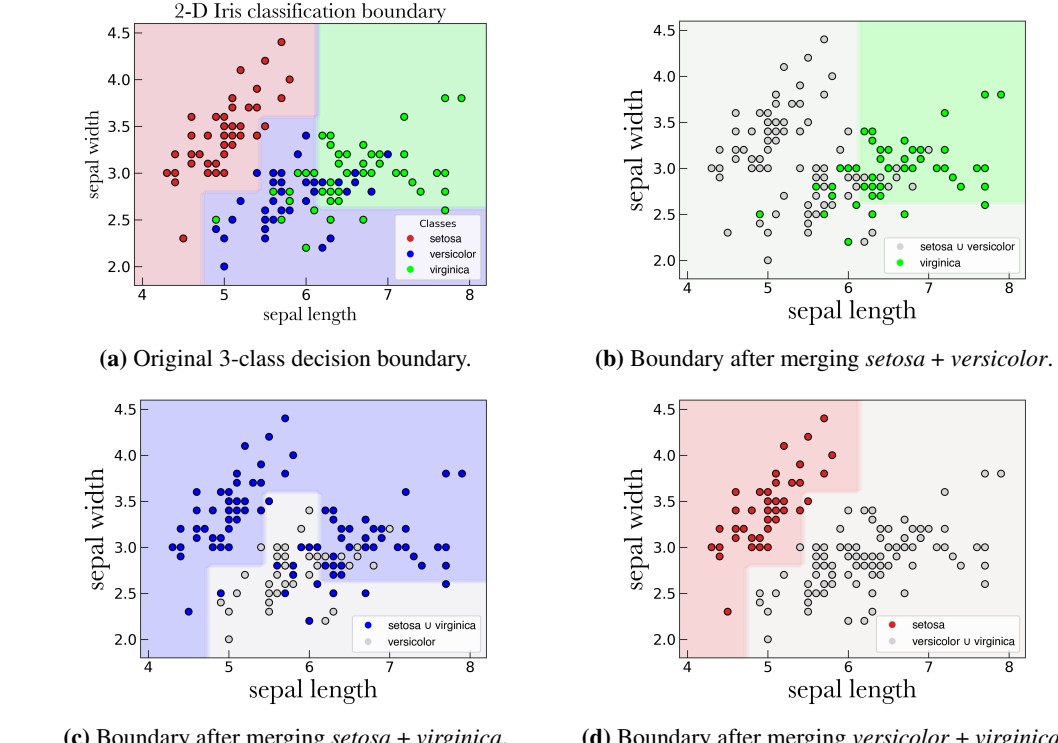

**(a)** Original 3-class decision boundary.

**(b)** Boundary after merging *setosa + versicolor*.

**(c)** Boundary after merging *setosa + virginica*.

**(d)** Boundary after merging *versicolor + virginica*.

Figure 1: (a) Original 3-class decision boundary on Iris sepal features. (b–d) Binary decision boundaries after merging two classes at a time: (b) setosa+versicolor vs. virginica, (c) setosa+virginica vs. versicolor, (d) versicolor+virginica vs. setosa. Points in merged classes shown in gray.

## 3 MULTI-CLASS FEATURE IMPORTANCE

We propose multi-class pairwise feature importance as the problem of providing feature importance separating a pair of classes in a multi-class classification scenario.

### 3.1 THE PAIRWISE FEATURE IMPORTANCE PROBLEM

Our goal is to provide discriminative information that is otherwise misrepresented or even absent. Some features may help discriminate two classes only, while otherwise not leveraged by the model. Hence, they might not be highlighted as important by global methods.

Consider the Iris dataset (Fisher, 1936), a simple yet popular multi-class classification task. It consists of 150 samples of iris flowers of species *versicolor*, *virginica* or *setosa*, with features *sepal length*, *sepal width*, *petal length* and *petal width*. For the sake of example, we here use only *sepal length* and *sepal width* (Fig. 1a). As can be seen, *versicolor* and *virginica* classes overlap, whereas *setosa* is linearly separable from the other two classes (Zaki & Meira, 2014). We fit a decision tree classifier on the simplified Iris dataset, using 50% of the data for training. Figure 1a shows the decision boundary. As expected, several *virginica* samples are misclassified as *versicolor* and multiple *versicolor* flowers are wrongly labeled as *virginica*. This testifies how the model does not effectively separate *versicolor* and *virginica*. On the contrary, no *setosa* sample is classified as *virginica* while just one is wrongly assigned to the *versicolor* class. A user interested in understanding this lack of separation, needs to get information on classes *virginica* and *versicolor* in Figure 1a.

To capture feature importance to discriminate two classes, we thus need to provide feature importance specfically for this pair of classes:

**Definition 1.** *Pairwise feature importance problem.*
*Given domain set $\mathcal{X}$, label set $\mathcal{Y}$, a black box classifier $h : \mathcal{X} \rightarrow \mathcal{Y}$, and two target classes*

$\sigma, \rho \mid \sigma \neq \rho$, *the pairwise feature importance problem is to find the feature importance of model $h$ as it pertains to separating the target classes $\sigma$ and $\rho$.*

Global FI methods, however, do not consider inter-class relationships, but provide importance scores across classes, and thus suffer from aggregation bias where importance not aligned with the global average is misrepresented.

We argue that FI methods solving the pairwise feature importance problem should meet the following requirements: it should (i) accept **any pair of class labels** as input, (ii) be **symmetric**, (iii) assign **zero to unimportant** features, and (iv) be **upper bounded**: To align with the above definition of the pairwise feature importance problem, stakeholders should be able to specify an **arbitrary pairwise** analysis. As in binary FI methods, but unlike e.g. GSHAP Bowen & Ungar (2020) (cf. Sect.2), feature importance should be **symmetric**, i.e. a feature score in separating class $\sigma$ from class $\rho$ equals the one for separating class $\rho$ from class $\sigma$. When a feature is **unimportant**, its importance should be 0, as intuition would suggest; as discussed as missingness in Lundberg & Lee (2017), features absent in some input should be attributed 0. Humans are able to reason better when dealing with a **limited range** of values than with potentially infinitely high values, hampering interpretability (Resnick et al., 2017; Jones et al., 2008; Pries et al., 2023; Adamczewski et al., 2020).

### 3.2 INTER-CLASS FEATURE IMPORTANCE (ICFI)

To address the above requirements, we propose ICFI, which adopts a strategy of removal-based explanations Covert et al. (2021), i.e., based on the principle of simulating feature removal to quantify feature influence. Concretely, we make use of permutation for removal. The underlying principle is that if a feature provides information about the target variable, breaking the association through permutation is reflected in model performance (Strobl et al., 2008). The feature is deemed unimportant when there is no significant increase in the empirical risk after permuting (Debeer & Strobl, 2020). A slight decrease in risk is also possible and is attributed to chance or to a sub-optimal model (Debeer & Strobl, 2020; Fisher et al., 2019):

For domain set $\mathcal{X}$, label set $\mathcal{Y}$, classifier $h : \mathcal{X} \to \mathcal{Y}$, loss function $l$, the true risk $R_t(h)$ is the expected loss of $h$ wrt. probability distribution $\mathcal{D}$ over $\mathcal{X} \times \mathcal{Y}$ (Shalev-Shwartz & Ben-David, 2014)

$$\mathcal{R}_t(h) = \mathbf{E}_{z \sim D}\left[l(h, z)\right] \tag{1}$$

Note that the true risk is not computable as the ML model has no access to $\mathcal{D}$ (Shalev-Shwartz & Ben-David, 2014), so we approximate it using the empirical risk $R(h)$, i.e., the average loss over a given data sample $(z_1, ..., z_N)$ (Shalev-Shwartz & Ben-David, 2014):

$$\mathcal{R}(h) = \frac{1}{N} \sum_{i=1}^{N} l(h, z_i) \tag{2}$$

For convenience, in the following we drop the $h$ in the risk's notation. We permute a feature by uniformly sampling one of its possible permutations. This means that if our data sample consists of $N$ records, each of the $N!$ permutations can be selected with probability $\frac{1}{N!}$. This corresponds to using the unconditional distribution as opposed the conditional one. The former is often used as an approximation for the latter Janzing et al. (2020). Whether in Chen et al. (2020) it is argued that which distribution to use depends on the application case, the authors in Janzing et al. (2020) argue that the sampling from the unconditional distribution is the better strategy.

The underlying intuition of ICFI is based on the idea that an increase in misclassifications between classes $\sigma$ and $\rho$ leads to a greater reduction in empirical risk when these two classes are merged. By combining the two classes into a single one, all previously misclassified instances between them are now classified correctly, thereby enhancing the overall performance of the model. We compute the decrease in empirical error when merging two classes through

$$\Delta \mathcal{R}^{\sigma \rho} = \mathcal{R} - \mathcal{R}^{\sigma \rho} \quad , \tag{3}$$

with $\mathcal{R}^{\sigma \rho}$ the empirical error when merging classes $\sigma$ and $\rho$. In Eq. 3, as in the remainder of the paper, we use Greek letters to refer to classes while we refer to features with Latin letters.

Eq. 3 would be incomplete without a clearer definition of the merging operation. Considering the case in which probabilities are provided by the model, when merging, two classes are considered as

one, resulting in a combined probability through summing. The merging definition can be trivially extended when models do not provide class probabilities by considering the class predicted having probability 1, while the remaining classes having probability 0. Regardless of the model, during evaluation, $\sigma$-labelled target data points are labelled as $\rho$. Merging is symmetric, i.e., merging $\sigma$ with $\rho$ is equivalent to merging $\rho$ with $\sigma$, making ICFI symmetric as desired.

If a model struggles to separate two classes, Eq. 3 will reflect it. To probe ICFI's intuition of relying on the empirical risk decreasing when merging two classes, we look back at the example displayed in Figure 1a. For the sake of the example, we use a mathematically simple loss function: the *zero-one* loss. The *zero-one* loss outputs 1 for a misclassified sample and 0 otherwise.

Dealing with three classes, three pairwise merges are considered, using all available samples. We merge *versicolor* with *virginica*, *setosa* with *versicolor* and *setosa* with *virginica*. The three scenarios are depicted in Figures 1b to 1d where points belonging to merged classes are coloured in gray and only the decision boundary between the two resulting classes is visible. The computed decreases in empirical error $\Delta R^{\sigma\rho}$ are respectively 0.20, 0.01 and 0.

As the decision boundary in Figure 1a indicates, the model struggles the most separating *versicolor* and *virginica*. $\Delta R^{\sigma\rho}$ reflects this, taking the highest value when merging *versicolor* with *virginica*. The model making no misclassifications between *setosa* and *virginica* is underscored by the null decrease in empirical error. Lastly, the only two misclassifications between *setosa* and *versicolor* cause a low 0.01 value of $\Delta \mathcal{R}^{\sigma\rho}$.

The examples in Figures 1b to 1d, thus show how the decrease in accuracy $\Delta \mathcal{R}^{\sigma\rho}$ captures model performance in separating classes. This makes $\Delta \mathcal{R}^{\sigma\rho}$ a key ICFI component. To evaluate feature importance for feature $j$, we permute $j$ and measure the difference in model performance. To evaluate ICFI we thus compute $\Delta R^{\sigma\rho}$ when permuting feature $j$ measuring

$$\Delta \tilde{\mathcal{R}}_j^{\sigma\rho} = \tilde{\mathcal{R}}_j - \tilde{\mathcal{R}}_j^{\sigma\rho} \quad , \tag{4}$$

with $\tilde{\mathcal{R}}_j^{\sigma\rho}$ being the empirical error when permuting feature $j$, and merging classes $\sigma$ and $\rho$. $\tilde{\mathcal{R}}_j$ denotes the empirical error after permuting feature $j$.

Eq. 3 and Eq. 4 evaluate decrease in empirical error respectively before and after permuting. ICFI could thus be defined taking the difference $\Delta \tilde{\mathcal{R}}_j^{\sigma\rho} - \Delta \mathcal{R}^{\sigma\rho}$ or through the ratio $\Delta \tilde{\mathcal{R}}_j^{\sigma\rho} / \Delta \mathcal{R}^{\sigma\rho}$. Both functional forms result in an unbounded target range, which goes against our specified requirements for interpretability. Moreover, setting an upper bound for the feature importance quantification, allows us to assess if a feature is highly important according to ICFI's definition. Without a limit on the importance evaluation, only assessments relative to other computed importances could be made.

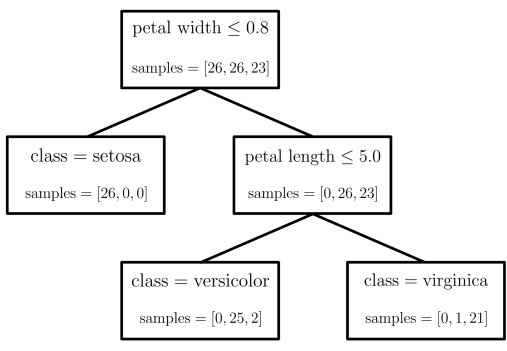

Figure 2: Decision tree of depth 2 classifying the Iris dataset. The *samples* entry indicate how many training samples for class *setosa*, *versicolor* and *virginica* respectively, go in a specific node.

We can get a bounded measure by constraining the target range between 0 and 1, with 0 signaling an unimportant feature. Starting from the ratio $\Delta \tilde{\mathcal{R}}_j^{\sigma\rho} / \Delta \mathcal{R}^{\sigma\rho}$, which has the advantage of normalizing the measure by the model performance before permutation, we need a function $f(x)$ mapping the output interval $[1, +\infty)$ to $[0, 1]$. $f$ should also be strictly increasing in order to preserve feature ranking. Choosing $f$ as: $f(x) = 1 - \frac{1}{x}$, would lead to defining $ICFI_j^{\sigma\rho}$ as $1 - \Delta \mathcal{R}^{\sigma\rho} / \Delta \tilde{\mathcal{R}}_j^{\sigma\rho}$. The problem with this definition is that actually, while $\tilde{\mathcal{R}}_j$ is smaller than $\mathcal{R}$ only in rare instances dictated by chance or by a sub-optimal model (Debeer & Strobl, 2020; Fisher et al., 2019), $\Delta \tilde{\mathcal{R}}_j^{\sigma\rho}$ can be smaller than $\Delta \mathcal{R}^{\sigma\rho}$ because the model distinguishes better the two classes after permutation. This would lead $1 - \Delta \mathcal{R}^{\sigma\rho} / \Delta \tilde{\mathcal{R}}_j^{\sigma\rho}$ to be negative. To see that $\Delta \tilde{\mathcal{R}}_j^{\sigma\rho}$ can be smaller than $\Delta \mathcal{R}^{\sigma\rho}$, consider the example in Fig. 2 with a decision tree on Iris data (Fisher, 1936) with all 4 features. The model perfectly classifies *setosa*, with the *setosa* leaf containing no *versicolor* or *virginica* samples

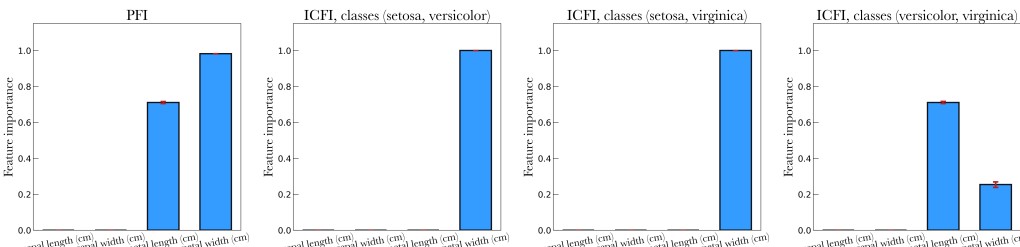

Figure 3: Global feature importance and ICFI for the model in Figure 2, fit on the *Iris* dataset. Global PFI (left) highlights the two features used; ICFI for *setosa–versicolor* (right) and *setosa–virginica* (center-right) shows reliance on the tree root (*petal width*), while ICFI for *versicolor–virginica* ranks *petal length* above *petal width* following the tree structure. Error bars are $95\%$ confidence interval; $y$-axes share the same range for easier comparison.

(*Samples* is the number of training samples for class *setosa*, *versicolor* and *virginica*, resp., in the node. ). Thus, in the setting where we do not permute any of the features, samples of *versicolor* or *virginica* are assigned to *versicolor* or *virginica* leaves; leaves where the model misclassifies some data, e.g. in the *versicolor* leaf, 2 samples are misclassified as *virginica*, but in the *setosa* leaf, no *versicolor* sample is misclassified as *virginica* and vice-versa. Now consider losing the information in the *petal width* feature by permuting it. More *virginica* and *versicolor* samples go to the left branch, as the model looses the discriminative power which separates *setosa* from *versicolor* and *virginica*. In the left branch, solely composed by the *setosa* leaf, the model does not misclassify *versicolor* with *virginica*, and merging *versicolor* with *virginica* leads to a low decrease in empirical error. Thus, $\Delta\tilde{\mathcal{R}}_j^{\sigma\rho}$ is smaller than $\Delta\mathcal{R}^{\sigma\rho}$. Hence, the interest is not in the difference $\Delta\tilde{\mathcal{R}}_j^{\sigma\rho} - \Delta\mathcal{R}^{\sigma\rho}$ but in its absolute value, i.e., $|\Delta\tilde{\mathcal{R}}_j^{\sigma\rho} - \Delta\mathcal{R}^{\sigma\rho}|$ which estimates by how much model performance differs after permutation:

**Definition 2.** *Inter-Class Feature Importance.*
*Given a black box classifier $h : \mathcal{X} \to \mathcal{Y}$, and two target classes $\sigma$, $\rho \,\big|\, \sigma \neq \rho$, the Inter-Class Feature Importance for some feature j is defined as*

$$ICFI_j^{\sigma\rho} = 1 - \frac{1}{1 + \left|\Delta\tilde{\mathcal{R}}_j^{\sigma\rho} - \Delta\mathcal{R}^{\sigma\rho}\right|/\Delta\mathcal{R}^{\sigma\rho}} \quad . \tag{5}$$

$ICFI_j^{\sigma\rho}$ quantifies the importance of feature $j$ in separating classes $\sigma$ and $\rho$, and meets the requirements of supporting arbitrary pairs of classes, being non-negative, zero for unimportant inputs, bounded, and symmetric wrt. the inspected classes (proof in App. A.5). Computational complexity is in line with existing permutation methods, but cheaper than e.g. SHAP, as outlined in App. B.

## 4    EXPERIMENTS

One of XAI's biggest challenges is its evaluation, due to lack of quantifiable metrics like accuracy. Furthermore, ground truth and evaluation standards are often missing (Molnar et al., 2023; Pries et al., 2023; Ali et al., 2023; Tritscher et al., 2023; Afchar et al., 2021; Adamczewski et al., 2020). Often, ground truth on feature importance actually refers to the data, not to the model.

Explanations are commonly assessed as a by-product of accuracy, or through case studies. Our experiments[1] utilize both assessment methods on real-world datasets. Different classifiers are used throughout the experiments to showcase the model-agnostic nature of ICFI. Note that this also means that we do not target model performance as such; instead, we focus on the quality of explanations of any model. Also in practice, explanations are required for models that perform as well as for those that do not. Unless stated otherwise, to compute ICFI, we use the cross-entropy loss, widely employed in classification tasks (Mao et al., 2023; Zhang & Sabuncu, 2018). ICFI can be applied with any other metric.

---

[1]Code available at `https://anonymous.4open.science/r/ICFI-5F1D/`

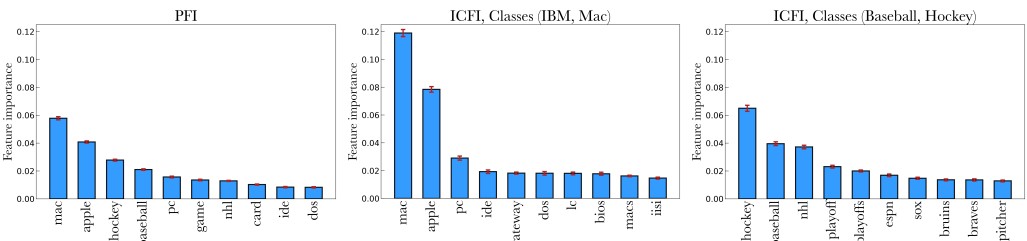

Figure 4: From left to right, PFI, ICFI of *IBM* and *Mac* and ICFI of *Hockey* and *Baseball*. The top ten ranked features are displayed. Pairwise similar classes, e.g., *Hockey* and *Baseball*, have highly discriminant features ranked on top. For the sake of comparison, plots are displayed with the same $y$ range. Error bars represent the 95% confidence interval.

In 4.1, an interpretable model is fitted to the Iris dataset (Fisher, 1936) as ground truth in feature importance rankings. We then test whether ICFI correctly quantifies ground truth feature importance. Section 4.2 exploits background knowledge in an NLP dataset to evaluate ICFI in a highly-dimensional setting. Words are used as features and some words are expected to be highly discriminative for a specific pair of classes, while are not supposed to be leveraged by the model in other pairs. We test if ICFI correctly captures this behavior. Our measure is compared to direct competitors in Section 4.3, to evaluate the quality of the feature importance rankings.

Across the experiments, we often offer PFI as a comparison, as it is the most popular feature importance measure which is both model-agnostic and inherently global. PFI provides a benchmark useful to understand the differences between our pairwise formulation and standard global FI.

### 4.1  3-CLASS DECISION TREE, NUMERICAL

The *Iris* dataset (Fisher, 1936) is a multi-class classification dataset. The task consists in distinguishing three different types of flowers i.e., *setosa*, *versicolor* and *virginica*, described by four features: *petal width*, *petal length*, *sepal width* and *sepal length*. We fit a CART decision tree (Breimann et al., 1984), setting the maximum depth at two as shown in Figure 2.

The decision tree has the advantage of being an inherently interpretable model. As discussed, we seldom have ground truth in the model and an interpretable one, together with a relatively simple dataset, offers the ground truth needed to evaluate ICFI. The tree structure in Figure 2, provides a feature importance ranking. First of all, the tree leverages just two features: *petal width* and *petal length*. *Sepal length* and *sepal width* should thus be labeled as unimportant. Moreover, features leveraged close to the root have a higher global influence than the

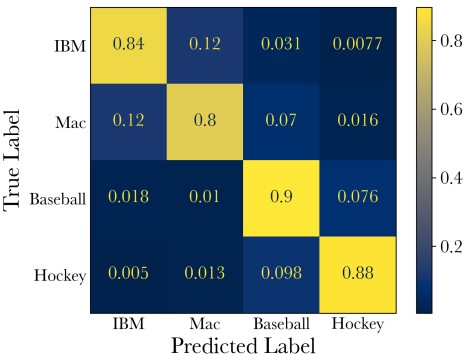

Figure 5: Newsgroup dataset normalized confusion matrix. The tiles showing the most misclassifications are the ones involving the *Hockey-Baseball* and *IBM-Mac* pairs.

ones used in lower nodes (Laugel et al., 2018). We thus expect *petal width*, used in the tree root, to be the globally most relevant feature. When distinguishing between *versicolor* and *virginica* instead, the model relies heavily on *petal length* as exemplified in Figure 2. Finally, when separating between *setosa* and the two classes to the right of the root, we foresee *petal length* to have low importance, as it cannot be used by the model to classify a sample as belonging to the *setosa* class.

Figure 3 shows the global PFI feature importance (left), the other three plots display ICFI for all three class combinations. Error bars represent the 95% confidence interval estimated through Bayesian inference exploiting Markov Chains Monte Carlo (MCMC); the strategy employed to compute confidence intervals in Figure 3, as well as in the rest of the paper, is outlined in Section A.3.

ICFI's rankings confirm the intuitions from the tree structure. First of all, *sepal width* and *sepal length*'s importance is negligible as it should be. On a global level, the feature which is at the root of the tree, *petal width*, is also the top ranked. Thanks to ICFI we can instead see how the

order is switched when the model tries to separate *versicolor* and *virginica*, in agreement with the structure showed in Figure 2. To separate *setosa* from the other two classes instead, the model mainly leverages *petal width*.

These traits of the model reasoning process are not captured by global feature importance methods like PFI, as they present generalized behavior and suffer from aggregation bias. Local methods instead fail to describe differences in two entire classes, but only one particular flower sample.

## 4.2 4-CLASS LOGISTIC REGRESSION, TEXT

20 newsgroup (Mitchell, 1999) is a text dataset containing newsgroup posts on 20 topics. For the purpose of this experiment, background knowledge provided by using words as features, allows us to consider four classes: *Hockey*, *Baseball*, *IBM* and *Mac*, chosen as they are pairwise similar and difficult to separate. The four classes total 2635 samples in the training set and 1573 in the test set. We encode words as features, resulting in a dataset with 4525 dimensions. Further details on the data pre-processing strategy are provided in Appendix A.4. The model is a binary logistic regression model fitted for each label. Figure 5 displays the model's confusion matrix computed on test data.

The tiles highlighting the most misclassifications are the ones involving the *Hockey-Baseball* and the *IBM-Mac* combinations, which are the pairings involving the most similar classes and thus, the most difficult to separate. The model performing better for certain class combinations than in others, raises the question of which are the important features for each pair.

Figure 4 shows PFI, $ICFI^{Baseball\text{-}Hockey}$ and $ICFI^{IBM\text{-}Mac}$, displaying the top ten ranked features. Global feature importance, as expected, highlights features relevant in both tasks, e.g., *mac*, *apple*, *hockey* and *baseball*, summing-up the whole model behavior. In both ICFI plots showed in Figure 4, we can instead see how words related to the inspected classes are the most important ones. For example, *mac* and *pc* are important features for the *Mac-IBM* class combination, while *nhl* and *pitcher* are within the highlighted features for *Baseball-Hockey*. Furthermore, the top spots are taken by features having high discriminant power between the classes of interest. Looking at $ICFI^{Baseball\text{-}Hockey}$ the first two ranked features are indeed *baseball* and *hockey*.

Furthermore, note that the feature importance values are relatively low w.r.t. the $[0, 1]$ range. This makes intuitively sense as the model has a high number of features to rely on. A measure with no upper bound couldn't have lead to such consideration, as we would not have any reference value to compare the computed FI with. ICFI correctly retrieves features used to separate an arbitrary pair of classes: a level of insight lost due to aggregation bias in global FI methods. We can indeed notice how, by looking at PFI's ranking, without background knowledge, we wouldn't be able to grasp which features are highly discriminative for which pair of classes.

## 4.3 COMPLEX MULTI-CLASS NEURAL NETWORK

We now consider model retraining, a common strategy to test the quality of feature importance rankings (Hooker et al., 2019; Meng et al., 2022; Sood & Craven, 2022). Conceptually, the better the feature ranking, the better a model trained with only its top $k$ most important features perform. The model to be explained and for which the features' rankings are created is a feed-forward neural network. As ICFI computes a ranking for a pair of classes for the model to explain, the retraining is carried out using a *One versus One*(OvO) strategy. A binary classification model is created for each class pair and fitted using ICFI's top $k$ features for each respective class pair. Each point is classified for each model and a final classification is obtained through a majority vote (Bishop, 2006). The *OvO* strategy is only employed in the retraining phase to evaluate ICFI, which is model-agnostic.

We study four real-world multi-class datasets: *Dry Bean* (mis, 2020), *Penguins* (LTER & Gorman, 2016), *Vehicle silhouettes* (Mowforth & Shepherd), and *Wine* (Aeberhard & Forina, 1991), which have 13611, 342, 423, and 178 samples respectively. *Dry Bean* is a classification dataset of grains belonging to 7 different varieties of dry beans. Each record has 16 numerical features describing the grain's shape and dimension. The *Penguins* dataset contains 4 numerical features about three different species of penguins. The goal of *Vehicle silhouttes* is to classify a given silhouette, leveraging 18 numerical features, as one of four types of vehicles. Lastly, *Wine* leverages the quantities of 13 wine constituents to label each record as belonging to three different cultivars.

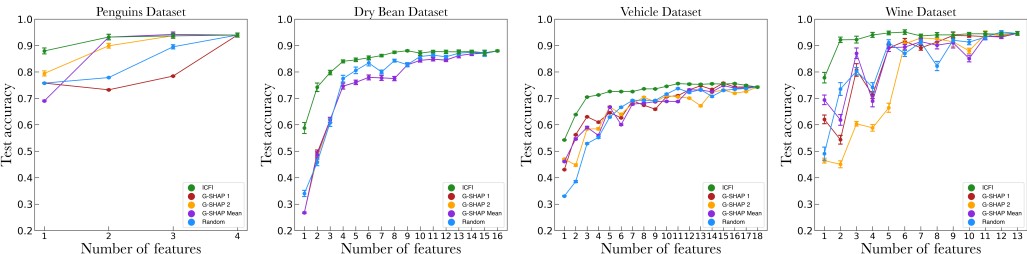

Figure 6: Test accuracy at different number of features selected. From left to right *Penguins*, *Dry Bean*, *Vehicle silhouettes* and *Wine* dataset. For the sake of better comparison the $y$ axis has the same range across all plots. Error bars show the $95\%$ confidence interval.

The model to explain is a feed-forward neural network, i.e., a black box model, which is trained for the multi-class classification problem at hand. ICFI is computed for each class pair and the top $k$ features for different values of $k$ are selected. For each value of $k$, a *One vs One* classifier is then fitted, i.e., a neural network is trained for each class pair, with the final decision obtained by aggregating models' outputs through a majority vote. Each neural network, binary classifying a class pair, will use the top $k$ features highlighted as most important by ICFI computed for that class pair. The better the features, the higher the performance of the retrained *One vs One* (Huang et al., 2020; Borisov et al., 2019).

We compare ICFI's rankings quality with other four feature selection strategies, which employ the same *One vs One* criteria in the retraining phase. We indeed compute GSHAP's feature importance, for each pair of classes, on the neural network we seek to explain, choosing the top $k$ features for binary models' retraining. GSHAP is not symmetric between the classes' pairs. Thus, three different comparison are generated by swapping the positive class between the two possible choices and by taking the average importance of the two possible settings. In the fourth benchmark, features are selected randomly for each class pair. Note that among all the methods computed in the evaluations, GSHAP variations which make use of SHAP values, are the most computationally demanding, as Shapley Values need to evaluate each feature coalition.

Figure 6 shows test accuracy at different $k$ values. Retraining based on ICFI consistently outperforms all GSHAP variations and the random strategy. Specifically, the increase in performance is most evident when few features are used, which is the most challenging setting and thus where the quality of the feature ranking matters most. Results indicate that features chosen with ICFI for each *One vs One* model carry more discriminative power than the features selected by competitors. ICFI can thus effectively find features with high discriminative power to separate two classes.

Due to lack of space, we refer to App. 4.3 for results showing that retraining global methods suffers from aggregation bias, failing to find most discriminative features for all pair of classes. Retraining of state-of-the-art FI methods like SHAP is outlined in App. A.2.

## 5    CONCLUSIONS AND FUTURE WORK

ICFI leads to more insights in explanations in a multi-class classification scenario, where current methods target binary classification only or suffer from aggregation bias. ICFI is more general as it quantifies feature importance in discriminating arbitrary pairs of classes. ICFI is a post-hoc, model-agnostic XAI method applicable to any existing ML model. ICFI relies on merging the two inspected classes and measuring the performance improvement to measure the model's performance in separating the pair of classes. We use permutation to mimic the absence of a feature, allowing us to quantify its importance. ICFI's output is bounded to a well-defined range making it easily interpretable to different stakeholders. The experimental evaluation demonstrates ICFI correctly retrieves features with high discriminative power for any pair of classes, aligning with ground truth. Competitors fail to to achieve comparable performance in model retraining. Future work includes alternative permutation strategies and feature attribution scores, leveraging our multi-class approach for a diverse set of XAI score approaches and different application domains.

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

# A APPENDIX

## A.1 MODEL RETRAINING, GLOBAL METHODS

As done in Section 4.3, we leverage model retraining to evaluate the quality of ICFI's explanations. The model to explain is the same as in Section 4.3, as well as ICFI's retraining strategy. Thus, a feed-forward neural network is used and a *One-vs-One* strategy is used in the retraining phase.

We compare ICFI's rankings quality with other four feature selection strategies. We indeed compute PFI, global SHAP and global LIME feature importance on the neural network we seek to explain, choosing the top $k$ features for model retraining. Note that among all the methods computed in the evaluations, SHAP is the most computationally demanding, as Shapley Values need to evaluate each feature coalition. In this scenario, each model in the *One vs One* classifier uses the same top $k$ features, as a global measure is used. In the fourth benchmark, features are selected randomly for each class pair. We include global methods in our comparison to show how the features important globally are not the most discriminative for each pair of classes.

Figure 7 shows test accuracy at different $k$ values. Retraining based on ICFI consistently outperforms retraining leveraging PFI, global SHAP, global LIME and when features are chosen randomly. Specifically, the increase in performance is most evident when a low number of features is used, which is the most challenging setting and thus were the quality of the features' ranking matter the most. This indicates how features chosen with ICFI for each *One vs One* model carry more discriminative power than the features selected by global XAI methods. ICFI can thus effectively find features with high discriminative power to separate two classes, not suffering from aggregation bias.

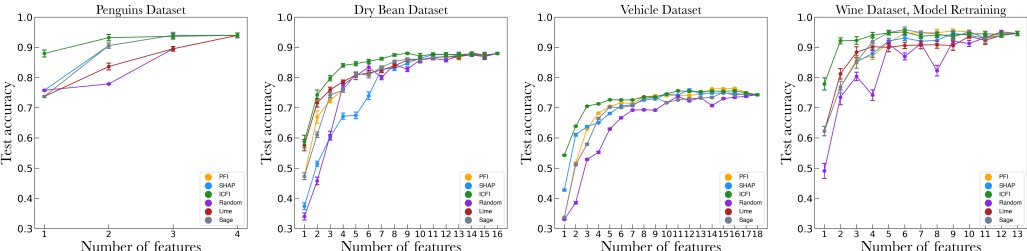

Figure 7: Test accuracy at different number of features selected. From left to right *Penguins*, *Dry Bean*, *Vehicle silhouettes* and *Wine* dataset. For the sake of better comparison the $y$ axis has the same range across all plots. Error bars show the $95\%$ confidence interval.

## A.2 MODEL RETRAINING, INTER-CLASS METHODS

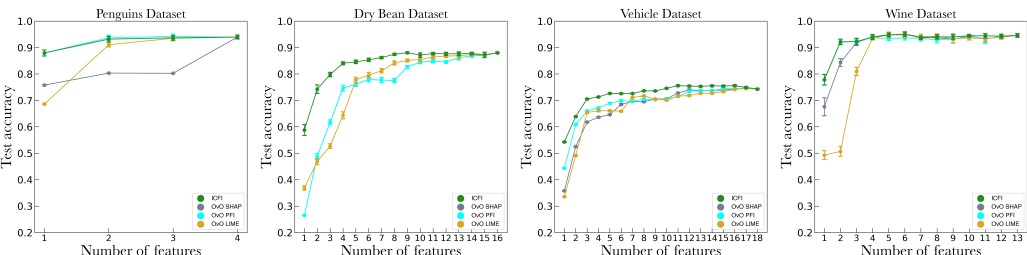

Figure 8: Test accuracy at different number of features selected. From left to right *Penguins*, *Dry Bean*, *Vehicle silhouettes* and *Wine* dataset. For the sake of better comparison the $y$ axis has the same range across all plots. Error bars show the $95\%$ confidence interval. OvO in the legend signals that the method has been used on each binary model of a *One vs One* approach.

In Section A.1, we benchmarked ICFI with global methods in order to assess how aggregation bias hides discriminant features for a pair of classes. Global methods do not have the same objective as ICFI because they explain the model globally without accounting for inter-class relationships. Here,

in order to allow these models to optimize for the same objective as ICFI, we use the following strategy. Instead of fitting and explaining a single neural network, we fit and explain multiple binary models, one for each class combination. We obtain a feature ranking for each pair of classes, using global SHAP, global LIME and PFI respectively, on each binary model. Retraining is performed analogously to the benchmarks in Section 4.3. The binary models can output a decision on the multi-class classification problem by aggregating each single models' prediction.

For a fair comparison with ICFI explanations already produced in Section 4.3, we use a comparable total number of parameters w.r.t. the neural network explained in Section 4.3. We note that while ICFI is completely model agnostic and handles multi-classification natively, state-of-the-art FI methods require, to find discriminative features for a pair of classes, a binary model for each class combination (i.e. a *One vs One* approach). This is computationally demanding and potentially sacrifices accuracy on the original task, on top of heavily constraining the classification strategy, compared to our earlier experiments. Note that if we would have run ICFI on the *OvO* model comprised of the multiple binary classifiers, this would still not explain the same model as the other FI methods because, they explain each binary model separately in this experiment.

Results are displayed in Figure 8, *OvO* in the legend signals that the XAI explanation has been computed on each binary model. ICFI is on par with PFI in the *Penguins* and *Wine* dataset while otherwise outperforming competing methods. ICFI achieves this without constraining in any way the classification strategy, not requiring a *One vs One* approach, demonstrating its novel contribution in multi-class classification.

## A.3 CONFIDENCE INTERVALS COMPUTATION

In each ICFI and PFI computation in the paper, the same strategy is employed to compute error bars, which relies on Bayesian inference. We run the feature importance computation $100$ times where each run differs because of the randomization in the permutation procedure. As an approximation, we assume the process is modelled by a Gaussian likelihood,

$$P(x|\mu, \sigma) = \frac{1}{2\pi\sigma^2} \cdot e^{-\frac{(x-\mu)^2}{2\sigma^2}} \tag{6}$$

We infer its mean $\mu$ and standard deviation $\sigma$ through a Bayesian approach. We make a conservative choice for both priors using a uniform distribution defined in the interval $[0, 1]$. We sample the posterior using a Markov Chain Monte Carlo (MacKay, 2003), which allows us to skip the evidence computation.

We generate chains using Python's *emcee* package (Foreman-Mackey et al., 2013). For each run we generate 20 chains with 6000 samples each, using the first 1000 as burn-in. The sampled points from each chain are then merged together. For our purpose, we consider only the $\mu$ parameter samples, quantifying feature importance with its mean and identifying the $95\%$ confidence interval excluding the first and last $2.5$ percentile of the distribution.

Figure 9 showcases one MCMC chain without burn-in (left) and the resulting $\mu$ distribution (right), for the *petal length* feature in the top left plot in Figure 9; i.e. for the PFI computation of the *petal length* feature in the *Iris* dataset.

In Figure 10, we show an example of the empirical distribution of ICFI for a specific feature. We see that the normal approximation for the likelihood is acceptable.

## A.4 20 NEWSGROUP PREPROCESSING

The data is preprocessed by creating a matrix representation of words count. A TD-IDF scheme (Baeza-Yates et al.) is then applied scaling down the impact of frequent tokens. Moreover, words appearing in more than half of the documents, or less than five times in total, are removed. This strategy allows us to rely on words as features, giving high interpretability. Crafting features with text embeddings, would instead imply features carrying less interpretability (Ribeiro et al., 2016).

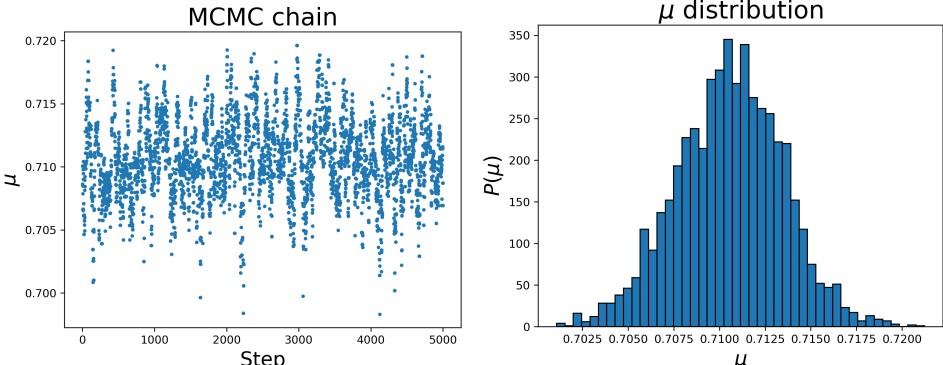

Figure 9: MCMC chain (left) and posterior marginal distribution of the $\mu$ parameter (right). The chain is run on the data obtained running $100$ times the PFI algorithm on the *petal length* feature of the *Iris* dataset.

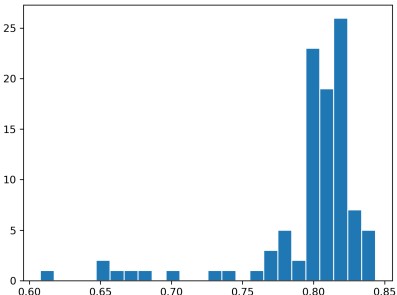

Figure 10: Empirical distribution of ICFI for a specific feature. We can see that the normal approximation for the likelihood is acceptable.

### A.5 ICFI PROPERTIES

The purpose of this section is to show that ICFI's definition in Eq. 5 implies a non-negative feature importance quantification which is bounded between $0$ and $1$ and a measure which is symmetric on the pair of classes. Note that boundedness between $0$ and $1$, implies non-negativity. We will then prove:

1. $ICFI_j^{\sigma\rho} \geq 0$  .

2. $ICFI_j^{\sigma\rho} \leq 1$  .

3. $ICFI_j^{\sigma\rho} = ICFI_j^{\rho\sigma}$   .

*Proof.*

1. We need to show that

$$\frac{\left| \Delta\tilde{\mathcal{R}}_j^{\sigma\rho} - \Delta\mathcal{R}^{\sigma\rho} \right|}{\Delta\mathcal{R}^{\sigma\rho}} \geq 0 \iff \Delta\mathcal{R}^{\sigma\rho} > 0 \quad . \tag{7}$$

It is left to prove that $\Delta R^{\sigma\rho} = \mathcal{R} - \mathcal{R}^{\sigma\rho} > 0$. If the relation is true for every single sample, it will consequently stay true when taking the average. The proof is shown for the cross-entropy loss: the scenario adopted in this paper's experiments and the most used loss in multi-class classification.

There are two possibilities to take into account:

- If the true label is neither $\sigma$ or $\rho$, $\tilde{\mathcal{R}}_j$ and $\mathcal{R}_j^{\tilde{\sigma\rho}}$ have the same value. The two original probabilities and the merged one are indeed multiplied by $0$ in the cross-entropy formulation.
- Without loss of generality, taking $\sigma$ as the true class, the cross entropy contribution for $\mathcal{R}$ is $-log\ p_\sigma$. With $p_\sigma$ the model probability for class $\sigma$. The merged one is instead $-log\ (p_\sigma + p_\rho)$ with the difference being

$$log\ (p_\sigma + p_\rho) - log\ p_\sigma =$$

$$= log\ \frac{p_\sigma + p_\rho}{p_\sigma} = log\ \left(1 + \frac{p_\rho}{p_\sigma}\right) > 0 \quad . \tag{8}$$

Note that in eq. 8 and in the cross-entropy computation, the probabilities are clipped avoiding $p_\rho$ and $p_\sigma$ to be exactly $0$.

2. To show that Eq. 5 always evaluates $\leq 1$ we need to show that

$$-\frac{1}{1 + \left|\Delta\tilde{\mathcal{R}}_j^{\sigma\rho} - \Delta R^{\sigma\rho}\right|/\Delta R^{\sigma\rho}} \leq 0 \quad , \tag{9}$$

which is immediate from Eq. 7.

3. The merge operation is completely symmetric and there are no operational differences in computing $ICFI^{\sigma\rho}$ and $ICFI^{\rho\sigma}$.

$\square$

## B    COMPUTATIONAL COMPLEXITY

XAI methods are often suffering from high computational complexity, and feature importance methods like SHAP often face challenges when scaled on high dimensional settings Das & Rad (2020).

Regarding ICFI, the proposed feature importance method averages the contribution of each instance and thus, has a linear dependency on the number of data points. The computation is independent for each feature so, a complete ranking on every feature scales linearly also with the number of features (and could be easily parallelized). We would like to point out that this is the same computational complexity as that of permutation strategies like PFI.

ICFI is computationally much cheaper than SHAP, as we do not need to evaluate every feature coalition, whose number scales exponentially with the number of features, but practically just one - for the permutation. To output a global explanation, LIME is computed for each instance and its complexity also depends on the surrogate interpretable model fitted for each single instance.

### B.1    RUN TIME ANALYSIS

GSHAP, as every method relying on Shapley Values' estimation, suffer from a higher computational complexity w.r.t. permutation methods **??**. To better highlight ICFI's computational advantages over Shapley Values' based explanation techniques, we provide a run time comparison between ICFI and GSHAP.

|  | Dry Bean | Penguins | Vehicles | Wine |
|---|---|---|---|---|
| **ICFI** | $65.0 \pm 0.6$ | $0.110 \pm 0.001$ | $1.123 \pm 0.007$ | $0.312 \pm 0.002$ |
| **GSHAP** | $982 \pm 8$ | $0.407 \pm 0.001$ | $6.84 \pm 0.01$ | $0.994 \pm 0.005$ |

Results show how ICFI is consistently faster than GSHAP with improvement scaling faster than linearly.

