# OpenReview forum: "ICFI: A Feature Importance Measure For Multi-Class Classification"
_ICLR.cc/2026/Conference — Submitted to ICLR 2026_

### Official Review · Reviewer_Qmjj · 2025-10-16

**Soundness:** 3
**Presentation:** 2
**Contribution:** 2
**Rating:** 4
**Confidence:** 5

**Summary:**

The paper introduces Inter‑Class Feature Importance (ICFI), a model‑agnostic, post‑hoc method for multi-class classification that quantifies how each feature helps discriminate a specific pair of classes. ICFI follows a removal‑based principle and uses feature permutation as a measure for feature importance: for a given feature, the method compares changes in empirical error between when the feature is permuted and when the feature is permuted and two target classes are merged. A feature is deemed important when its permutation increases the error of separating the selected classes. Empirically, the method identifies discriminative features for chosen class pairs and shows advantages over baselines.

**Strengths:**

* The problem of pairwise FI in multiclass classification is clearly defined, an perspective missing from many existing FI tools.
* The method is model‑agnostic and post‑hoc: applicable to arbitrary black‑box classifiers without retraining the base model.
* The proposed measure is intuitive, simple and meet the requirements for an effective FI method.
* Empirical evidence is presented to demonstrate that ICFI‑derived features are more meaningful and accurate than baselines method.

**Weaknesses:**

1. **Novelty**: The claim that multi-class explanations are not addressed elsewhere is too strong. For example, Vo et al. [1] present a model‑agnostic, post‑hoc multiclass FI method. The authors are encouraged to further reflect the differences between the two works.

2.  **Computational cost**: Computing importance requires multiple model evaluations per feature and per class pair, which may be burdensome for large deep models.

3. **Extension to other modalities and large-scale classifiers**: It is unclear how the approach extends to high‑dimensional, less interpretable modalities (e.g., images or videos) as well as other classification tasks such as questions answering where the loss function is not simply cross-entropy loss. In this case, the measure in Def. 2 may no longer satisfy the requirements.

[1] Vo et al. An Additive Instance-Wise Approach to Multi-class Model Interpretation (ICLR’23)

**Questions:**

1. Why is permutation used to represent feature removal, instead of common approaches like feature dropping, zero/mean substitution? How sensitive are results to different removal strategies?

2. For images or texts, permutation may break semantics of the input content. Could the authors comment on this issue and how it affects the quality of output features from the proposed method?

3. How would ICFI be adapted for images/videos or pairs of modalities e.g., text-image pairs or text-video pairs? In today's AI landscape, I think it is important to reflect on how the proposed framework would be used for these modern black-box models.

4. How stable are pairwise importance scores across different random seeds, sample sizes, and class imbalance conditions?

---

> ### Author Response · Authors · 2025-11-20
>
> We are thankful to the reviewers for the insightful commentary. We appreciate our paper being described as tackling a perspective missing from many existing FI tools. Furthermore we thank for the comments stating that the proposal is intuitive and effective.
>
> **W1**:
> The work by Vo et al. does not have the same objective as ICFI: instead, it learns local explanations, and while it focuses on multi-class problems, it does not study inter-class relationships. Our claim is not that there is no multi-class classification explanation method but, none that  explain inter-class relationships. The only other method with that object is GSHAP, which is clearly outperformed by ICFI.
>
> **W2**:
> Computational complexity is a common problem for XAI, ICFI has the same computational complexity as popular methods like PFI and lower computational complexity than widely employed ones like SHAP.
>
> **W3**:
> ICFI is best suited for tabular data. As for other methods like PFI and SHAP, ICFI can be applied to every data type once the right features are chosen. For example, a common way to apply FI on images (apart from heatmaps that are typically generated by non-model-agnostic methods because they require the model to be differentiable) is via creating features as super-pixels. Using such features is an interesting direction for future work for ICFI, which could then be extended to other such modalities.
>
> ICFI is generalisable to other loss functions and does not need to be  restricted to the cross-entropy loss. We will add a comment in this regard to our revised version of the paper. As far as the properties are concerned, symmetry is dependent on merging definitions only (and thus robust across losses). Boundedness between 0 and 1 is satisfied for any typically used  loss function, as the important step in ICFI is the merging operation, which can only reduce the number of misclassifications, and thus reduces empirical risk.
>
> **Q1**:
> Feature dropping in the context of FI would require training a separate model without the feature in input. This creates two problems: first, a computational one, i.e. this would require  training the whole model from scratch, and second, a conceptual one: when evaluating the model without the feature, this corresponds to querying a different model (namely, one without access to this feature) which would hinder its ability to explain the original model. Mean imputation is considered deprecated as it is not neutral, i.e.,  not removing the dependency between the feature and the output, but comparing with the mean as input which has unknown importance in the model. As a consequence, choosing a value like the mean might favour some class or inaccurately assess the importance of the feature more generally speaking.
>
> **Q2**:
> We propose and study ICFI for tabular data mainly. Indeed, other modalities would require some extension in terms of semantics. Here, existing strategies are likely immediately applicable. Image data could  follow the same strategy as e.g. SHAP and LIME use for images (e.g. super pixels) with similar downsides.
>
>
> **Q3**:
> In general, ICFI can be applied to any modality once the feature to permute is established, e.g. a token in textual data or in text-image data. Future work could study inter-class importance specifically for inputs like images or text or the combination of the two.
>
>
>
> **Q4**:
> In the paper we compute confidence intervals through Bayesian-based methods (described in the appendix). Specifically, MCMCs help capture variability with iterations having different random seeds. Computed confidence intervals show that ICFI improvements are not due to random chance but are statistically significant.

---

> ### Comment · Reviewer_Qmjj · 2025-11-25
>
> Thank you for the responses.
>
> 1/ Regarding W1, if one interprets an "inter-class" explanation as to find the features that distinguish between the classes (as said so in your Def. 1), then the explanations provided by Vo et al. also satisfy this purpose, by identifying features that distinguish one class from the others. Another reason why I raised this point is that your paper title clearly indicates "multi-class explanation", rather explicitly stating "inter-class" as you would like to highlight. The authors are not expected to compare your method with Vo et al. empirically. I mentioned the paper only to say that the setting of "multi-class explanation" can be interpreted differently.
>
> &nbsp;
>
> 2/ Regarding permutation strategy, I am not fully convinced by the justification against feature removal. Removal-based explanation is a well-studied problem with plenty of solutions, from either theoretical or empirical perspectives or even both.  I agree with the authors that mean substitution is not a proper way, but there are many other substitution schemes or removal strategies like marginalization (see Covert et al., 2021).
>
> Yes, training surrogate models is also one way (among many!). Training the model from scratch is obviously not ideal, but if this gives meaningful explanations, then I don't think it is a major issue.
>
> While I understand the concern that *"it would hinder the ability to explain the original model"*, I view it as a challenge to address, rather than the reason for choosing an alternative strategy. I personally think, if done properly, feature removal is a more appropriate and natural mechanism to evaluate the effect of a feature - rather than permutation, from which I feel that the feature importance may get entangled with the bias effect from feature ordering.
>
> Covert, I., Lundberg, S., & Lee, S. I. (2021). Explaining by removing: A unified framework for model explanation. Journal of Machine Learning Research, 22(209), 1-90.
>
> &nbsp;
>
> 3/ However, the concern about permutation is not the main reason for my rating. For the chosen problem, the paper indeed provides a solid solution. The method is an extension of feature importance from single-class to inter-class setting, which however to me is rather an incremental contribution. The method is limited to tabular data, while prior works have already done explanations on texts or images, though possibly not fully effective. Given the current AI landscape, I think the bar needs to be raised and I believe more advanced explanation methods, for example for more complex modalities, would be more beneficial to the community.
>
> For the above reasons. I retain my evaluation.

---

> ### Author Response · Authors · 2025-11-25
>
> Thank you for taking the time to to further discuss our paper, below are our further clarifications
>
> 1) Definition 1 clarifies that the goal of our work is to differentiate between two target classes, i.e., NOT between one class and the rest, but specific pairs of classes. This is the major difference to existing work. Indeed, it is thus not only multi-class explanation, but also the inter-class aspect that are of concern here. Please note, though, that in case of binary classification, the two coincide, so multi-class classification is the setting in which the difference becomes important.
>
> 2) Thank you also for pointing to the work by Cover et al. which provides a nice overview over different ways of removing features - notably including also permutation (emphasising also the value and computational benefit of permutation approaches). Please note that our answer to question 1 is not arguing against feature removal, but against feature dropping (surrogate models in Covert et al.) - as we seem to agree, feature dropping requires retraining with the mentioned computational and conceptual downsides. Feature removal on the other hand has the benefit of not requiring retraining and explaining the original model. Removal using zero or mean substitution is not necessarily “neutral”, which constitutes a major challenge. For example, it has been shown that depending on the task, the choice of “neutral” has major impact on feature importance, leading to potentially highly unreliable results (e.g. Z. Fernando, J. Singh, A. Anand: A study on the Interpretability of Neural Retrieval Models using DeepSHAP.  SIGIR ’19), as you already acknowledged for mean substitution. Regarding marginalisation, as analysed in Covert et al. our permutation strategy is a form of marginalisation over the joint distribution of data and target, which is argued by Janzing et al. (Janzing et al. Feature relevance quantification in explainable AI: A causal problem AISTATS 2020) to be superior to marginalising over the conditional distribution of the data given the target. We thus adopt a removal strategy that relies on permutation that also Covert et al. consider beneficial in their overview.
>
> 3) While our method indeed adopts permutation as the strategy for feature removal, its contribution also lays in the novel strategy that offers an assessment of feature importance for pairs of classes without the need of retraining a complex multi-class classifier. Our paper demonstrates that for the important case of tabular data, this is an effective and highly efficient approach. As the paper also shows, in Section 4.2., it is also very effective for text. It can be straightforwardly extended to other data modalities such as images using existing feature concepts, so there is no limitation to our method in this regard.

---

### Official Review · Reviewer_GThT · 2025-10-27

**Soundness:** 2
**Presentation:** 3
**Contribution:** 2
**Rating:** 2
**Confidence:** 4

**Summary:**

The paper proposes a novel feature importance method for multi-class classification. The method tackles the problem of providing not only one set of feature importances, but one for each pair of classes, thus offering more insight into the classifier. The method is based on the idea of observing decrease in empirical risk when two classes are combined, in combination with permutation-based feature importance. Some experiments are provided showing that the method gives sensible results and outperforms GSHAP adapted to the same problem.

**Strengths:**

Overall, the paper is well-written and most of the ideas are clearly communicated and easy to understand. Feature importance is a very saturated field, but this work tackles a novel subproblem. I like the basic idea of combining classes.

**Weaknesses:**

The proposed method is relatively simple and does not bring any extremely innovative methodology or theoretical results, which is nothing wrong by itself, but the I would expect a very strong empirical evaluation or (even better) a practical use-case that demonstrates not only that the method works but that the problem of requiring additional insights into (pairwise) relationships between classes is really a problem in need of a solution.

The current experiments do not convince me (see Questions).  As the authors also say, evaluation of XAI is a big challenge and there doesn't seem to be any shortcut to a sound empirical evaluation (https://icml.cc/virtual/2025/poster/40169). The first two experiments establish that there is nothing clearly wrong with the method, which is OK. The retraining experiment and comparison with GSHAP I do not understand. If the goal of the method is to provide insights into how the model classifies, then this is far from a realistic assessment (yes, it is common to do this in XAI/ML papers, but it doesn't make it any less unrealistic). Also, it seems to me that GSHAP was forced into this comparison, not being a method developed for the same purpose. I might be wrong, but the paper doesn't do a good job of describing exactly what GSHAP is or how it was adapted.

And I might have other issues with the paper on things that I currently don't quite understand and/or were not explained clearly enough (also see Questions).

Minor comments:
- Some extra effort seems to have gone into squeezing this to fit the page limit (Figure 6 caption has no space to breathe, etc.).
- ).One
- The proposed method operates on model risk not on model predictions directly. So, technically, it is not explaining what the model does, but what features contribute to the models predictive performance. Often the same, but not always.

**Questions:**

Q1:  Finally, why not include some global feature importance into the comparison? The problem of masking the feature importance of globally less important features that are important for certain pairs of classes might be exaggerated. I'd imagine that for a low number of classes the global ordering would be decent (definitely better than random).

Q2. Computational complexity: First, it would really help if the computational complexity is stated more explicitly, instead of "in line with existing permutation methods but cheaper than SHAP". Second, I'm not convinced that the latter is correct. The proposed method requires for each feature a constant number of permutations and each permutation requires a model prediction? Any decent implementation of SHAP should also be linear in the number of features and will contain the model prediction (you don't go through all subsets of coalitions).

Q3. Permutation importance has certain failure cases, compared to SHAP, for example. Why not combine the idea of combining two classes but then use Shapley values instead of permutation importance?

Q4.  I'd remove the explicit "Definition 1" from definition of the pairwise feature importance problem. It is not necessary and it is not precise. Informally we would probably agree on what "as it pertains to separating the target classes $\sigma$ and $\rho$" means, but what does it really mean? A model never trully 100% focuses on separating only two classes (unless there are only two classes).

Q5. The interval computation in A.3 seems like overkill. The (Bayesian posterior) mean and standard deviation of a process where 100 independent samples are given is estimated using Markov Chain Monte Carlo? Unless I'm missing something, the only possible justification would be that we use uniform priors on the two parameters and therefore can't use the analytical solution. But if we are going to be so precise as to not allow values outside of [0,1] then why use a Gaussian likelihood, which is clearly not appropriate. Burn-in also doesn't make sense (why not just pick a sensible starting value, like the empirical mean and standard deviation). To summarize, average +/- 1.96 * standard deviation of the sample / sqrt(100) should give essentially the same results.

Q6. I'm unsure about the upper/lower bound requirement. First, the requirements, as stated, would allow for a method that assigns arbitrarily low negative feature importances (we require irrelevant features to have 0 and to have an upper bound; there is nothing saying that a relevant feature can't have a negative importance, for example, if it decreases predictive performance). I'll assume that the intention was for them to be bounded between 0 and an upper bound (which might as well be 1). I'm not convinced by the argument that people prefer bounded things therefore bounding is better. That is, it is mathematically easy to bound things, but with it we change the scale of the feature importance. Are these importances even comparable across class pairs for same risk? Are they comparable across different risks?

---

> ### Author Response · Authors · 2025-11-20
>
> We are grateful for the reviewer’s constructive feedback and provide our clarifications below.
>
> **W1**: Our evaluation shows that pairwise feature importance uncovers information that global methods miss, evident in the first three experiments, where aggregation bias hides inter-class insights, and in the retraining experiment (appendix), where global methods fail to match ICFI’s accuracy. Among existing approaches, only GSHAP explicitly targets pairwise class comparisons; as the authors note in their Iris example, they seek “features which distinguish Versicolour from Setosa,” aligning with ICFI’s goal. We will clarify this shared scope in the revised version. Although no other method pursues this objective, we still compare against global methods and include additional methods crafted to share ICFI’s goal, all reported in the appendix.
>
> **W2**: The goal of the retraining experiment is to evaluate the quality of the importance rankings. The central assumption is that a good ranking identifies the features carrying the most predictive information according to the model; therefore, training on only these features should yield strong performance. This also indirectly evaluates whether the explanation captures how the model actually classifies. If an FI method is flawed and highlights features unrelated to the output, then retraining on those features will perform poorly. Thus, this experiment provides a technical assessment of the information the FI method extracts from the model, which we will clarify in the revised version. While correlated features can mean that some unused features remain informative for retraining, making this evaluation imperfect, consistently strong retraining performance across datasets and methods (including global methods we added in the appendix) supports that the explanation reflects the model’s reasoning. Finally, we highlight that the first two experiments already show that ICFI works and that global methods fail due to aggregation bias.
>
> **Q1**: Due to space, comparison with global methods, as well as other methods crafted to have the same objective as ICFI, are benchmarked in the appendix (reference in the main text at line 470). ICFI shows superior performances across the board.
>
> **Q2**: In the submitted version, computational complexity is described briefly in the appendix. We  will further expand this discussion in the updated version. SHAP approximation exists, and while it does  not have to evaluate all feature coalitions, it has to evaluate more than one. ICFI, on the other hand, evaluates just one. Our expanded run time analysis in the revised version of the paper (using state-of-the-art approximations), clarifies how ICFI is significantly faster than SHAP. Furthermore, note that permuting one feature can be seen as the most extreme Shapley Value approximation, evaluating only one coalition, and exactly corresponds to the most extreme approximation of Shapley Values described in (Štrumbelj et al. KAIS 2014).
>
> **Q3**: Shapley values for pair-wise scoring is indeed an interesting direction for future work. It would offer a different trade-off between computational complexity and axioms. In our work, we have focused on an efficient solution that empirically shows high quality results.
>
> **Q4**: Thank you for noticing that, we will upload a further revised version of the paper with a more precise definition.
>
> **Q5**: We begin with burn-in. Chains were generated with emcee (Daniel et al., PASP 2013) using 20 walkers. Initialising all walkers at the same point would stall the ensemble sampler, so we use different starting points around the empirical mean. As these points are biased, keeping the full chain without burn-in is inappropriate; using only post-convergence samples is the most sensible choice. A simpler single-chain MCMC might also have worked, but we used MCMC to ensure robustness. ICFI users do not need MCMC, as confidence intervals are only part of the experimental evaluation. Regarding priors, empirical FI values lie in [0,1]. A broader prior (e.g., an improper uniform prior) would still work but slow convergence and lengthen burn-in. Supported by the central limit theorem, FI values are well-approximated as normally distributed. In the revised version, we will add empirical FI histograms showing this approximate normality and justifying the likelihood choice.
>
> **Q6**: Normalising, and thus bounding, changes the scale of feature importance, but the original unbounded scale is hard to interpret. A raw value like 75 has no clear meaning, whereas a value such as 0.8 on a [0,1] scale is far more intuitive (Miller et al., Artif. Intell. 2019). We do not claim that importances are comparable across risks, but boundedness helps relate them to the specific risk at hand. For example, a top feature with importance 0.95 (rather than 0.1) more clearly signals its potential impact. Overall, bounded importances ease interpretation beyond simple relative rankings.

---

> > ### Comment · Reviewer_GThT · 2025-11-25
> >
> > Thanks to the authors for taking the time to address all the comments. Based on the authors’ response, I am increasing my score. As it stands, I see the work as adequately addressing a problem in feature importance that might have some potential application but is likely to be extremely niche. In terms of theoretical contribution or demonstrated practical utility it does not particularly stand out from the dozens of novel feature contribution methods proposed every year that never see any practical application. I do view the field of feature contributions (XAI in general) as one where the focus should ultimately be the user.
> >
> > **Q2 (cont.):** I’m not arguing that ICFI is not significantly faster that at least any Shapley Value approximation that I’m aware of, but I don’t think we can make the claim that it is computationally less complex (in the conventional sense, in orders of magnitude).
> > (Štrumbelj et al. KAIS 2014) is an example of a method that needs $k$ model predictions per feature. $k$ may be large, but it does not grow with problem size (it depends on the variability of outputs, which is bounded). So, the computational complexity is “number of features times computational complexity of a prediction”. I’m assuming that is also the computational complexity of ICFI? It’s not just the computational complexity of a single prediction?
> > I think it is important to clarify this, because it will help us understand how the methods scale and if ICFI scales an order of magnitude better or just a constant better, which does benefit us on smaller datasets, but would diminish as the number of features grows.
> >
> > **Q6 (cont.):** Not that this is a critical point of the paper, but I disagree that it is clear that there is an inherent practical benefit of bounded scales (Miller et al. doesn’t support or make any explicit claims in that regard). If a raw value of 75 has no meaning, then neither will its scaling by some relatively arbitrary and noisy value. Furthermore, 0.9 in one task is not comparable with 0.9 in another task, and we can’t really say that 0.9 is “twice as important as 0.45” in the same task either (or that 0.1 and 0.2 are the same distance apart as 0.8 and 0.9), so we have an ordinal scale? Not to say that people don’t feel more comfortable with bounded values, just that it is far from clear that there is an actual benefit beyond that.

---

> > > ### Author Response · Authors · 2025-11-25
> > >
> > > Thank you for engaging in the discussion and for the further clarifications.
> > >
> > > **1)** ICFI can find valuable applications in all multi-class settings. Very often in multi-class settings some pairs of classes are more similar than others and/or making sure that there are no misclassifications between a specific pair is more important than for other pairs of classes. The fact that ICFI can be applied in all these multi-class scenarios make it, in our opinion, not a niche contribution and one that can find very solid practical applications e.g. in model validation. ICFI stands out as it is the first apt and efficient approach that provides explanations on the difference of any pair of classes in multi-class settings. This has important practical implications, as existing approaches often handle only binary classification, or ignore the fact that class differences may be obfuscated by other classes.
> > >
> > > **2)** Thank you for clarifying that. As you note, ICFI’s complexity is indeed “number of features times computational complexity of a prediction”. This makes ICFI’s complexity better than exact SHAP. As the run-time analysis added in the paper shows, the constant factor over SHAP approximations, leads to large discrepancies in run-time.
> > >
> > > **3)** We agree that boundedness alone does not suffice for comparability across classes. Still, as you note, people tend to find it easier to interpret if the scale of values is bounded. While it does not mean that values can be interpreted as stand-alone values, it is a notable qualitative advantage in practice.

---

### Official Review · Reviewer_92cg · 2025-10-31

**Soundness:** 2
**Presentation:** 2
**Contribution:** 2
**Rating:** 4
**Confidence:** 3

**Summary:**

The authors propose ICFI, a global feature importance measure centred on multi-class classification. The proposed approach is a model-agnostic, post-hoc approach that computes feature importance by considering the “pairwise feature importance” problem, this is achieved by evaluating how a feature contributes to the separation of classes $\sigma$ and $\rho$. ICFI offers an elegant and computationally efficient means of assessing feature importance.

**Strengths:**

•	The related works / references are quite thorough.
•	The idea is interesting and conceptually simple.
•	The experiments favour ICFI.
•	It is to my understanding that this can be applied to any method and does not require any separate model or model retraining.

**Weaknesses:**

•	Whilst interesting conceptually, the main contribution seems to be merging classes – and the FI measure, while the measure is intuitive, it lacks in axiomatic grounding.
•	I think computational runtime is a strength when compared to GSHAP (SHAP in general), a clear table of runtime – which is not presented as a discussion would be ideal.
•	The comparison is quite limited. There is no benchmarking against other methods such as Expected Gradients (EG), Integrated Gradients (IG), Manifold IG, Layer-wise Relevance Propagation (LRP), SmoothGrad, and other methods which have seen wide application since SHAP for neural network-based experimentation. Libraries such as Captum would help provide a broader set of benchmarks. As well as global feature attribution methods such as SAGE [1].
•	Considering the above, it is also notable that GSHAP is an arXiv paper.

[1] Ian C. Covert, Scott Lundberg, and Su-In Lee. 2020. Understanding global feature contributions with additive importance measures. In Proceedings of the 34th International Conference on Neural Information Processing Systems (NIPS '20). Curran Associates Inc., Red Hook, NY, USA, Article 1444, 17212–17223.

**Questions:**

•	Is there a benefit to the proposed permutation approach as opposed to sampling from a background/reference dataset? –  this strategy employed in methods such as EG (an extension of IG).
•	The manuscript proves boundedness and (more centrally) symmetry but does not analyse standard attribution axioms (e.g., completeness/efficiency, monotonicity, linearity). Please (a) state explicitly which axioms ICFI satisfies or violates, (b) discuss practical implications of any violations, and (c) motivate why ICFI’s objective (global pairwise discriminative power) warrants these trade-offs. It is worth considering if there exists a global FI analogue. (I refer the authors to [1])
•	If computational or conceptual mismatch prevents some suggested baselines, please can the authors explain?

---

> ### Author Response · Authors · 2025-11-20
>
> We thank the reviewer for the suggestions provided as well as for characterising our idea as interesting. In the follwing we hope to properly reply to questions.
>
> **W1**:
> Axiomatic grounding: the class of permutation methods, which ICFI is a part of, trade-off Shapley Value axioms (e.g. additivity) for computational efficiency, but retain missingness. Effectively, ICFI needs to compute one feature coalition only, i.e. the one where the feature inspected is removed. Evaluating all coalitions would make ICFI satisfy Shapley Values’ axioms at the cost of a bigger computational effort. In practice, ICFI’s explanations are of high quality at much better computational performance.
>
> **W2**:
> Thank you for the suggestion, we have provided a table of runtime in the updated version of the paper, Appendix B.1.
>
> **W3**:
> ICFI is model-agnostic, whereas EG, IG, Manifold IG, LRP, SmoothGrad are not. One of ICFI strengths is that it can be applied to any model while these methods need a differentiable model. Regarding global methods, as mentioned briefly in the main text at line 470, we offer a comparison in the appendix, including other baselines constructed to have the same objective as ICFI. Here, global methods show worse performance than ICFI: including SHAP, LIME, and PFI (SHAP and LIME global explanations are obtained by averaging local ones). Following the suggestion of the reviewer, we added SAGE to that comparison in the updated version of the paper. Our new experimental results show that SAGE  performs similarly to other global methods and is outperformed by ICFI. Specifically, ICFI better performs when only a small number of features are available for training: the most challenging setting and thus the one benefitting the most from high-quality explanations. Captum’s implementations, other than the global methods already benchmarked (LIME, SHAP, PFI), almost exclusively have non-model-agnostic gradient-based methods.
>
>
>
>
> **W4**:
> While GSHAP is an arxiv paper, we include it because it is the only work that has a similar objective as ICFI: model-agnostic inter-class explanations. Comparison with further state-of-the-art global methods and specialized  baselines  in the appendix show that ICFI outperforms all methods in terms of retraining performance (briefly mentioned in the main text in line 469-470).
>
> **Q1**:
> It is actually a strong advantage of permutation approaches like ICFI that we do not need a background dataset. Such background datasets can be difficult to establish and can bias FI attribution if not well designed  as well as produce misleading explanations [1]. Furthermore, permutation preserves the marginal distribution, providing the right notion of dropping features [2].
>
> **Q2**:
> Regarding the axioms respected by SHAP, ICFI satisfies the missingness axiom and trades-off the rest for computational efficiency. For ICFI to respect Shapley Values’ axioms (like GSHAP which does not have good results) we would have to evaluate all feature coalitions to compute each feature’s contribution to the decrease in empirical error. This would keep ICFI’s intuition of evaluating the decrease in empirical error after merging intact, and change how the contribution to that value from each feature is computed. This could be interesting future work, but we expect a major computational cost and poor scalability. Experimental results show that permutation-based feature removal leads to high-quality explanations.
> Please note that SAGE does not have the same objective as ICFI. SAGE computes Shapley Values on the loss of the model (similar to LossSHAP but directed toward global explanations). Its main difference to aggregating local SHAP explanations is how it approximates Shapley Values, and the fact that Shapley Values are not estimated on the model output but on the model loss. Hence, SAGE is not able to provide feature importance for separating a pair of classes, but only across all classes.  In the updated version of the paper, we add SAGE to the other global approaches already benchmarked in the appendix. Results are in line with other global methods with ICFI outperforming them especially when a low number of features are at disposal.
>
> **Q3**:
> As mentioned above, global feature importance methods are benchmarked in the appendix. Even though they do not provide the same type of explanations as  ICFI, i.e. inter-class feature importance, this comparison shows clearly that their aggregation bias may obfuscate inter-class behaviour.
>
> [1] Aas, Kjersti, Martin Jullum, and Anders Løland. "Explaining individual predictions when features are dependent: More accurate approximations to Shapley values." Artificial Intelligence 298 (2021): 103502.
> [2] Janzing, Dominik, Lenon Minorics, and Patrick Blöbaum. "Feature relevance quantification in explainable AI: A causal problem." International Conference on artificial intelligence and statistics. PMLR, 2020.

---

### Meta-Review · Area_Chair_coJc · 2025-12-27

**Summary:**

This submission proposes a model-agnostic feature-importance measure for multi-class classification that outputs pairwise feature importances. The proposal combines a class-merging idea with permutation-based feature removal, and measures changes in empirical risk. Reviewers agree the method is intuitive and the paper is generally well written, but the consensus is that the contribution is incremental in the feature-importance space and is not supported by sufficiently strong theory or compelling evaluation. Key concerns include limited axiomatic grounding, unclear computational advantages, reliance on permutation with known failure modes, and the lack of a convincing practical use case or broader validation beyond tabular settings. The rebuttal adds some comparisons (e.g., runtime table, inclusion of SAGE) and clarifies positioning, but several core issues remain.

**Reviewer Concerns:**

Partially addressed:
- Added runtime table and analysis, more baseline comparisons including SAGE and global methods.
- Clarified ICFI’s goal (pairwise class separation) and which axioms are traded off for efficiency.

Still outstanding:
- Incremental novelty
- Permutation-removal limitations not fully resolved
- Limited scope beyond tabular

**Reviewer Scores:**

Reviewer GThT: Initially 2; likely moves to borderline (e.g., 4).

Reviewer 92cg: likely stays around 4. Might increase to 6.

Reviewer Qmjj: unlikely to change.

---

### Decision · Program_Chairs · 2026-01-26

Reject